# Investigation of Physical and Mechanical Characteristics of Rubber Materials Exposed to High-Pressure Hydrogen

**DOI:** 10.3390/polym14112233

**Published:** 2022-05-31

**Authors:** Sang Koo Jeon, Jae Kap Jung, Nak Kwan Chung, Un Bong Baek, Seung Hoon Nahm

**Affiliations:** Center for Energy Materials Metrology, Korea Research Institute of Standards and Science, Daejeon 305-340, Korea; sangku39@kriss.re.kr (S.K.J.); jkjung@kriss.re.kr (J.K.J.); nk.chung@kriss.re.kr (N.K.C.); ubbaek@kriss.re.kr (U.B.B.)

**Keywords:** rubber, high-pressure hydrogen, volume ratio, hydrogen contents, tensile strength

## Abstract

Rubber materials play a key role in preventing hydrogen gas leakage in high-pressure hydrogen facilities. Therefore, it is necessary to investigate rubber materials exposed to high-pressure hydrogen to ensure operational safety. In this study, permeation, volume swelling, hydrogen content, and mechanical characteristics of acrylonitrile butadiene rubber (NBR), ethylene propylene diene monomer (EPDM), and fluorocarbon (FKM) samples exposed to pressures of 35 and 70 MPa were investigated. The results showed that the volume recovery and hydrogen desorption behavior of EPDM with the highest permeation were fast whereas those of FKM with the lowest permeation were slow. The volume of NBR with the highest hydrogen content expanded after decompression. In contrast, FKM swelled the most despite having the lowest hydrogen content. After exposure to high-pressure hydrogen, the compression set (CS) slightly increased due to internal cracks, but the tensile strength decreased significantly with increasing pressure despite the absence of cracks in the fracture area of all tensile specimens. It was concluded that the decrease in tensile strength is closely related to the volume increase because of the relationship between the relative true strength and the volume ratio.

## 1. Introduction

Global environmental issues, such as pollution, depletion of fossil fuels, and global warming have necessitated the development of alternative energy sources. Hydrogen has emerged as a good candidate for an alternative source of energy source. However, the energy density of hydrogen must be increased to ensure its efficient use as an energy source. Methods such as liquefying hydrogen at low temperatures or compressing hydrogen at high pressures have been used to enhance the energy density. High-pressure hydrogen facilities commonly use metal pipes, high-pressure storage vessels, and sealing materials to prevent leaks. The ductility of most metals is reduced under a high-pressure hydrogen environment, which enhances several risk factors. High-pressure hydrogen degrades the mechanical properties of metals, such as tensile strength, and elongation, and results in a relative reduction of area [1,2,3,4]. Moreover, the development of fractures is relatively faster under a high-pressure hydrogen environment, which reduces the life expectancy of the materials [5,6,7,8,9]. A wide variety of studies have been performed to evaluate the effects of high-pressure hydrogen on metals, and efforts are being made globally to ensure the safety of high-pressure hydrogen facilities. However, research on the effects of high-pressure hydrogen on sealing materials has only recently begun. Yamabe et al. [10,11] fabricated NBR and EPDM with various amount of carbon black and silica to examine their penetration properties and hydrogen content under the effect of high-pressure hydrogen. In this study, transparent rubber without fillers was also used to observe the development of blisters and cavities in real-time. The internal damage was enhanced with increasing hydrogen pressure, and the internal cavities grew and eventually caused fractures and cracks both inside and outside the rubber samples upon repeated exposure to high-pressure hydrogen [12,13,14]. Additionally, the crack growth properties of various types of rubber were investigated in high-pressure hydrogen environments [15]. To examine the deterioration mechanism of rubber under high-pressure hydrogen, Fujiwara et al. [16,17] evaluated the changes in the chemical structure of NBR after exposure to hydrogen at 100 MPa using nuclear magnetic resonance (NMR) and infrared spectroscopy (IR), and reported that the chemical structure did not change. The fracture behavior of fabricated EPDM O-ring under compressive strain was observed, and it was reported that the O-ring fractured upon repeated exposure to high-pressure hydrogen [18].

It is important to evaluate the physical properties of rubber under compressive strain because rubber is subjected to compressive strain in high-pressure hydrogen facilities. Compression set and compressive stress relaxation tests can be conducted to evaluate the compressive properties of rubber materials, and their service life and durability have been studied using these methods [19,20,21,22,23,24,25]. The properties of rubber in a high-pressure hydrogen environment determine the safety and lifetime of hydrogen facilities because rubber materials suffer damage, such as cracks and fractures when exposed to high-pressure hydrogen. Therefore, it is necessary to evaluate the effect of high-pressure hydrogen on the compression properties of rubber. Menon et al. [26,27] performed compression set tests on sealing materials exposed to high-pressure hydrogen, and reported that the durability of rubber was reduced by high-pressure hydrogen. In that study, micro-computed tomography (CT) was also used to examine the inside of the rubber samples and microvoids and microcracks were found. Various additives are used in the production of the rubber materials, and the type and amount of additives often determine the physical properties of the rubber. The physical properties of rubber exposed to high-pressure hydrogen differ according to the type of additive used. Previous studies have mainly focused on evaluating the effects of high-pressure hydrogen on the type and content of fillers [10,11,12,13,14,15,16,17]. However, Simmons et al. [28] conducted a study using helium ion microscopy and TEM and reported that plasticizers dissolved in rubber, with and without fillers, were extracted to the surface when exposed to high-pressure hydrogen. Thus, they determined that this phenomenon of phase separation was one of the factors causing changes in the rubber properties.

To ensure the safety of high-pressure hydrogen facilities, it is necessary to evaluate the effect of high-pressure hydrogen on rubber materials. In this study, the effects of hydrogen on the physical and mechanical characteristics of NBR, EPDM, and FKM were investigated. The hydrogen permeability test was conducted on non-exposed rubber materials but the volume swelling, hydrogen content, and mechanical tests were performed after exposure to hydrogen pressure of 35 and 70 MPa. Then, any change in characteristics and differences in the effects of the high pressure were compared between the rubber materials.

## 2. Experimental

### 2.1. Materials

NBR (KNB 35 L with an acrylonitrile content of 34%) and EPDM (KEP2320 with an ethylene content of 58%) were produced by the Kumho Petrochemical (Seoul, Korea) company and FKM (Viton A601 C) was purchased from the Pyunghwa Chemical company (Daegu, Korea). Fast extruding furnace carbon black (N550, Orion Engineered Carbons, Grand Duchy of Luxembourg) was used to reinforce the properties of NBR and EPDM, and medium thermal carbon black (N990, Orion Engineered Carbons, Grand Duchy of Luxembourg) was mixed into FKM. NBR and EPDM formed a crosslinked structure with sulfur and peroxide, and the compounds were prepared using various rubber additives, such as vulcanization accelerators, co-crosslinking agents, and plasticizers. The compound formulas are presented in Table 1 and the structure of NBR, EPDM, and FKM is shown in Figure 1. The NBR, EPDM, and FKM vulcanizates were provided in the shape of plates and cylinders by a domestic company in Korea and the manufacturing process was omitted due to the company confidentiality.

### 2.2. Hydrogen Charge

All specimen vulcanizates were exposed to high-pressure hydrogen to evaluate whether they were affected by high-pressure hydrogen except for the permeability test. The specimens were placed in a high-pressure hydrogen vessel with up to 120 MPa, and then a purge was carried out three times to remove any impurities inside the vessel using vacuum and hydrogen gas at 5 MPa. After finishing the purge, hydrogen was pressurized to 35 and 70 MPa in the high-pressure hydrogen vessel because these pressures are the maximum pressure for the high-pressure hydrogen tank currently loaded in commercial the fuel cell electric vehicle. The specimens were exposed to each pressure for 24 h, and then the vessel was rapidly decompressed to atmospheric pressure.

### 2.3. Permeation Test

The hydrogen permeability test for each specimen was conducted according to ISO 15105 using differential pressure methods [29]. Figure 2 shows a schematic of the permeability system, which consists of an upper and a lower chamber. The plate specimen was 45 mm in diameter and 2 mm thick. A metal filter was installed to prevent deflection of the specimen due to the pressure difference, and a filter paper was inserted under the specimen to determine the area through which hydrogen permeated. After the specimen was set up between the upper and lower chambers, both chambers were evacuated to approximately 1 × 10^−3^ Pa. While maintaining the vacuum state in the lower chamber, the upper chamber was pressurized to 102 KPa with hydrogen. Hydrogen gas permeated through the specimen from the upper to the lower chamber. The pressure in the lower chamber gradually increases owing to the permeating hydrogen gas, and was measured with a pressure gauge until the rate of pressure increase became constant.

### 2.4. Volume Swelling and Hydrogen Content Test

The volume of the specimen (12 mm in diameter and 2 mm in thickness) before and after exposure to high-pressure hydrogen was calculated by measuring its thickness and diameter using a laser micrometer (LSM-6200, Mitutoyo, Japan). After the exposure to high-pressure hydrogen, the volume of the specimen was measured according to the elapsed time. Three specimens were tested to obtain reliable results, and the relative change in volume was calculated using the following equation:(1)$$\mathrm{Relative}\;\mathrm{change}\;\mathrm{in}\;\mathrm{volume}\;\left(\%\right)=\frac{V-V_{0}}{V_{0}}\times 100$$
where, *V*_0_ is the volume before exposure to high-pressure hydrogen and *V* is the volume after exposure to high-pressure hydrogen.

A specimen (50 mm in diameter and 2 mm in thickness) was used to measure the dissolved hydrogen content in each rubber. Five specimens were weighted using a microbalance before and after exposure to high-pressure hydrogen [30]. After decompression, the weight of the specimens was measured according to the elapsed time, and the residual hydrogen content was calculated using the following equations:(2)$$\mathrm{Residual}\;\mathrm{hydrogen}\;\mathrm{content}\;\left(\mathrm{wt}.\mathrm{ppm}\right)=\frac{W-W_{0}}{W_{0}}\times 1,000,000$$
where, *W*_0_ is the weight before exposure to high-pressure hydrogen and *W* is the weight after exposure to high-pressure hydrogen.

### 2.5. Mechanical Test

Compression set (CS) provides useful information about the recovery performance of samples and was performed according to the ASTM D395 standard [31]. The cylinder specimen (29 mm in diameter and 12.5 mm in thickness) was compressed at 25% using the equipment composed of two plates and spacers and this equipment was heated to 100 °C for 72 h in oven (OF-02, JEIO Tech, Daejeon, Korea). After finishing the heat treatment, the samples were removed from the equipment, and cooled for 30 min to room temperature. The CS value is defined by the following equation:(3)$$\mathrm{CS}\;\left(\%\right)=\frac{h_{o}-h_{t}}{h_{o}-h_{s}}\times 100$$
where, *h_o_* and *h_t_* are the heights of the original sample and tested sample, respectively. *h_s_* is the height of the spacer used to compress the sample by 25%.

The tensile tests on dumbbell-shaped specimens (ASTM C type) were carried out according to the ASTM D 412 standard [32]. The specimens were punched out from about 2 mm thick disc plates and tensile specimens were exposed to high-pressure hydrogen. All tests were conducted using an MTS machine with a 5 kN load cell, and crosshead speed was 500 mm/min with a gauge length of 25 mm (MTS criterion model 42, MTS, Eden Prairie, MN, USA).

### 2.6. Scanning Electron Microscope (SEM) Analysis

The cut section in the longitudinal CS specimen and the fracture surface of the tensile specimen were observed using SEM (Jeol 7100F, JEOL Korea Ltd., Seoul, Korea). After completing the tests, each specimen was analyzed to compare the differences between before and after exposure to high-pressure hydrogen.

## 3. Results and Discussion

### 3.1. Permeation of Hydrogen

Figure 3 exhibits the permeability results for NBR, EPDM, and FKM. Hydrogen gas was injected into the upper chamber under vacuum at time zero. The pressure in the upper chamber increased with the injection of hydrogen gas, and then hydrogen gas permeated through the rubber specimen toward the lower chamber at a relatively low pressure. Finally, the pressure in the lower chamber increased due to the permeation of hydrogen gas. In Figure 3, it can be observed that the pressure in the lower chamber started to increase within a short time after the injection of hydrogen into EPDM, whereas the pressure started to increase last for FKM. Additionally, the pressure increased rapidly for EPDM, but gradually increased for FKM. Hydrogen gas permeating through the rubber specimen is constantly released and the increase of pressure becomes constant. The slope of the straight line of the section in which the pressure change was constant was fitted, as shown in Figure 3. The time lag, which is the point at which the fitted line meets the initial pressure line, was obtained, and the diffusivity coefficient (D) can be determined by the following equation:(4)$$D=\frac{d^{2}}{6\theta }$$
where *θ* is the time lag, and *d* is the thickness of the specimen. The permeability coefficient (P) can be obtained as follows:(5)$$P=\frac{V_{c}}{\mathrm{RT}p_{h}A}\times \frac{dp}{dt}$$
where T and R are the temperature and gas constant, respectively. A and *d* are the area and thickness of the specimen. *V_c_* is the volume of the lower chamber, *p_h_* is the pressure of hydrogen in the upper chamber, and *dp/dt* is the slope in Figure 3. Using the diffusivity (D), permeability (P), and solubility coefficient (S) can be calculated according to the following equation:(6)P=D×S

Table 2 shows the permeability, diffusivity, and solubility coefficient for NBR, EDPM, and FKM, and all coefficients are average values. The permeability and diffusivity coefficients of NBR and FKM were lower than those of EPDM. This means that NBR and FKM have excellent permeation resistance to hydrogen gas and EPDM has low sealing performance compared with the others because hydrogen gas is able to permeate through it easily. The permeability coefficient of rubber materials with respect to hydrogen gas is generally affected by many factors, such as the polarity of the material, filler type, various additives, and the type and size of the gas molecule [33]. Moreover, the permeability coefficient of hydrogen gas decreases with increasing filler content [34]. It was considered that the effect of the filler content was negligible because the amount of carbon black filler contained in all specimens is almost the same and the permeation characteristics mainly depended on the type of rubber material. When the material and gas have different polarities, diffusion is reduced in these materials [35]. NBR and FKM are polar rubber but hydrogen is nonpolar gas, resulting in a low diffusivity coefficient, whereas EPDM, which is nonpolar, has a higher diffusivity coefficient for hydrogen gas. The permeability coefficient of EPDM was also remarkably higher than those of NBR and FKM. While the diffusivity coefficient has a large influence on the permeability coefficient, the solubility coefficient has little effect under the same gas conditions [36]. These results coincide with those of previous studies, in which the permeation characteristics of rubber materials were evaluated using various methods, and the solubility coefficient of FKM with a relatively high density was calculated to be the lowest because of a lack of space in which hydrogen can be dissolved compared with NBR, which has a low density and different fillers [30,37]. This relationship between density and permeation has been observed for polymer materials, and the diffusivity decreases with an increase in density [10,38]. In addition, the solubility coefficient is affected by the specific area of carbon black filler because of the high interaction between the filler and hydrogen gas [10]. MT carbon black filler in FKM has a large specific area compared to FEF carbon black filler in NBR and EPDM. Accordingly, a large amount of hydrogen gas can be adsorbed to the FEF carbon black filler than the MT carbon black filler, and the solubility coefficient of NBR and EPDM was higher than that of FKM. Also, the solubility coefficient of NBR was higher than EPDM because the hydrogen gas and NBR have different polarities and the diffusion and permeation of the hydrogen are slow in NBR. Consequently, hydrogen gas rapidly diffused in EPDM, which has the same polarity as hydrogen gas, but NBR and FKM, which have different polarities and dense structures, inhibited the diffusion and permeation of hydrogen gas.

### 3.2. Hydrogen Contents and Swelling

Figure 4 shows the results of the volume change and residual hydrogen content over time for NBR, EPDM, and FKM exposed to high-pressure hydrogen at 35 and 70 MPa after decompression and the fitted line based on the measured data. It is evident that the volume of all rubbers exposed to high-pressure hydrogen swelled, and that the volume expanded further when the pressure of hydrogen was increased. As shown in Figure 4a, after the beginning of the measurement, the volume decreased with time regardless of the type of rubber material. However, the rate of the volume reduction was different for each material. The expanded volume of EPDM quickly decreased to the original volume, whereas NBR and FKM showed a tendency for a relatively slow volume decrease. In particular, the volume of FKM swelled the most, and its rate of volume reduction was the slowest compared with other materials. It can be confirmed that the volume of FKM exposed to high-pressure hydrogen at 70 MPa was maintained in a swollen state of more than 60% until at least 8 h had elapsed after decompression. The residual hydrogen content was measured at all times after decompression and the results are shown in Figure 4b. At the beginning of the measurement, the residual hydrogen content was at its highest, but it gradually decreased over time similar to the volume recovery results. In addition, the residual hydrogen content increased for higher hydrogen pressures, regardless of the type of rubber material. These results show that the number of hydrogen molecules that can penetrate the rubber material increases as the hydrogen pressure increases and that they are desorbed from the inside of the rubber materials after decompression. However, the desorption behaviors of hydrogen for each rubber are different. The residual hydrogen content of NBR and EPDM dramatically decreased in the beginning and then showed a gradual decreasing tendency. However, in the FKM analysis, a slightly different tendency was observed and the overall residual hydrogen content relatively decreased slowly. Moreover, the total dissolved hydrogen content in FKM was the lowest compared with that in NBR and EPDM. From these results, it can be determined that the volume of the rubber materials swelled with the absorption of hydrogen upon exposure to high-pressure hydrogen, and the swollen volume returned to the original state for all rubbers with the desorption of hydrogen after decompression. Additionally, the degree of swelling and the hydrogen content depend on the hydrogen pressure on the rubber materials.

Figure 5 compares the behavior of the volume recovery and the desorption of hydrogen for NBR, EPDM, and FKM over a long time using the fitted line in Figure 4, and the vertical axis is expressed in logarithmic form. The volume decreased linearly until it returned to the original size, whereas the residual hydrogen content rapidly decreased at the initial time and then gradually decreased after a certain period. Thus, the volume recovery and desorption of hydrogen are different after decompression. It is shown in Figure 5 that even though the volume of EPDM returned to its original value, the hydrogen adsorbed inside was not completely desorbed, and the remaining hydrogen content was not zero. Nishimura et al. also found that the volume of NBR with a carbon black filler exposed to high-pressure hydrogen could be recovered after decompression but the hydrogen was still present inside the material [39]. For hydrogen desorption from the rubber materials with a carbon black filler, the hydrogen adsorbed to the matrix and to the filler exhibited different desorption behavior, and the hydrogen adsorbed onto the carbon black filler was very slowly desorbed [40]. Therefore, as shown in Figure 5, the rapid decrease in the early stage indicates the mechanism by which hydrogen desorbs from the matrix, afterwards, the moderate decrease is attributed to desorption from the fillers. The times to recover 1% of the volume of NBR, EPDM, and FKM exposed to 35 MPa were estimated to be approximately 16, 4, and 51 h, respectively. In addition, the time to reach 10 wt.ppm of NBR and EPDM was 22 and 14 h, respectively, but the time for FKM was very long, estimated to be 83 h. In terms of both the volume recovery and desorption of hydrogen, FKM required a considerable amount of time, while for EPDM it was relatively short. This result is consistent with the results of the permeability test. It has been demonstrated that the desorption behavior of hydrogen is closely related to the permeation behavior [40]. However, the relationship between volume recovery and the permeation behavior is unclear. Because the volume decreases linearly, and the desorption behavior of hydrogen has two slopes, it is assumed that the time to return to the original state before exposure to high-pressure hydrogen is different.

### 3.3. Mechanical Characteristics

When the CS specimen was exposed to high-pressure hydrogen, the volume significantly swelled and bubbles were non-uniformly formed on the surface [41]. Therefore, the CS test was conducted after the volume of the specimen had been recovered. Figure 6 presents the results of CS for NBR, EPDM, and FKM exposed to hydrogen pressure of 35 and 70 MPa. Regarding the rubber materials that were not exposed to high-pressure hydrogen, the CS value of FKM is very low and that of EPDM is very high. This means that FKM has a superior resilience but EPDM is vulnerable to the applied temperature and compression. Generally, the average CS value of the rubber materials exposed to high-pressure hydrogen showed a slightly increasing trend but those values changed within the error range. Thus, the change of CS characteristics by high-pressure hydrogen was not clear. However, Simmons et al. [42] investigated the change in CS value after exposure to high-pressure hydrogen and they reported that the characteristic of CS in NBR without a filler and plasticizer was deteriorated after exposure to high-pressure hydrogen, whereas the performance of CS in EPDM without a filler and plasticizer was improved. Also, the CS value of NBR with a filler and plasticizer was considerably increased, whereas the CS value of EPDM with a filler and plasticizer was slightly changed after exposure to high-pressure hydrogen. From these results, the change in CS value by high-pressure hydrogen did not show a constant trend, and the reason for the change in CS value was not analyzed.

SEM analysis was performed to investigate the damage inside the rubber materials caused by high-pressure hydrogen. After the CS specimen exposed to hydrogen pressures of 35 and 70 MPa was cut in the longitudinal direction, the cut section was observed using SEM, as shown in Figure 7. Small cracks were observed in the cut section of NBR exposed to 35 MPa hydrogen pressure, whereas both large and small cracks were observed in the cutting section of NBR exposed to 70 MPa hydrogen pressure. When the EPDM was exposed to hydrogen pressures of 35 and 70 MPa, there was no difference because both main cracks occurred inside. The FKM was not damaged by a hydrogen pressure of 35 MPa, but both large and small cracks were observed at hydrogen pressures of 70 MPa. Although damages clearly occurred inside the rubber materials by exposure to high-pressure hydrogen, the change in CS vales was not clearly distinguished. Therefore, this means that the evaluation of the effect of high-pressure hydrogen through the CS value can be inappropriate based on our and previous results.

Figure 8 presents the nominal stress-strain curves of NBR, EPDM, and FKM before and after exposure to hydrogen pressures of 35 and 70 MPa and 1 h after decompression because it takes time to remove the tensile specimen from the high-pressure hydrogen vessel after decompression. The curves of the samples not exposed to high-pressure hydrogen were initially consistent with the curves of those exposed to high-pressure hydrogen, but the curves of the tensile specimen exposed to high-pressure hydrogen exhibited a different behavior in terms of a weakening in the stiffness and also became weaker with an increase in hydrogen pressure. Thus, both tensile strength and fracture elongation were decreased by exposure to high-pressure hydrogen, and the tensile characteristics deteriorated as the pressure increased because the tensile characteristics upon exposure to 70 MPa hydrogen pressure showed a greater decrease than after exposure to 35 MPa hydrogen pressure.

The tensile tests were performed for NBR, EPDM, and FKM exposed to 35 and 70 MPa hydrogen pressure over time after decompression, and the results of the relative changes in tensile strength and volume are shown in Figure 9. The section marked with a slanted line indicates the range of tensile strength of the specimens not exposed to high-pressure hydrogen, and the relative changes in tensile strength and volume were calculated as follows:(7)$$\mathrm{Relative}\;\mathrm{change}\;\mathrm{in}\;\mathrm{tensile}\;\mathrm{strength}\;\left(\%\right)=\frac{T_{o}-T}{T_{o}}\times 100$$
(8)$$\mathrm{Relative}\;\mathrm{change}\;\mathrm{in}\;\mathrm{volume}\;\left(\%\right)=\frac{V_{o}}{V}\times 100$$
where *T_o_* and *T* are the tensile strengths before and after exposure to high-pressure hydrogen, respectively, and *V_o_* and *V* are the volumes of the tensile specimens before and after exposure to high-pressure hydrogen, respectively.

An increase in the relative change in tensile strength means that the tensile strength decreased, and a decrease in the relative change in volume indicates volume swelling due to exposure to high-pressure hydrogen. The relative change in tensile strength substantially increased at 1 h after decompression for all materials. As time elapsed, the tensile strength showed a tendency to recover, and then the tensile strength finally returned to its original value. The relative change in volume decreased at 1 h after decompression, and then the volume of the tensile specimen finally returned to its original value, similar to that of the tensile strength. The tensile strength and volume of EPDM were almost recovered, but those of FKM had recovered only slightly at 5 h after decompression. Hence, the velocity at which the tensile strength and volume recovery differ according to the type of rubber material but it was confirmed that the tendencies for volume recovery and tensile strength recovery were consistent regardless of the hydrogen pressure.

Figure 10 presents the fracture areas of the tensile specimens not exposed to high-pressure hydrogen and exposed to a hydrogen pressure of 70 MPa. The fracture area of the specimen exposed to a 35 MPa hydrogen pressure was excluded because the tensile strength of the specimen exposed to a 70 MPa hydrogen pressure was significantly decreased. Compared with the fracture area before and after exposure to high-pressure hydrogen, blisters and inner cracks were not observed, and there was no difference. Nishimura et al. reported that blisters and cracks were formed on the fracture area of tensile specimens without fillers, but the defects were not observed for specimens with fillers after exposure to high-pressure hydrogen [43]. However, the tensile strength of all rubber materials also decreased, regardless of the presence of a filler, and these results are consistent with our results. It is interesting that defects such as blisters and cracks were only formed inside the CS specimen but not in the tensile specimen. To be able to make the defects, the hydrogen molecules must be sufficiently agglomerated inside the rubber materials [10]. Because of a thicker and larger volume of the rubbers compared with the tensile specimen, it is difficult for solute hydrogen molecules to escape from the inside to the outside of the CS specimen, so that hydrogen molecules can easily aggregate. However, the creation of internal defects in the tensile specimen is harder because the hydrogen molecules can quickly escape in the thickness direction before they are able to aggregate.

### 3.4. Discussion

It is difficult to accurately measure the volume swelling and hydrogen content of rubber materials immediately after decompression because of the time required to remove the specimens. Therefore, the volume swelling and hydrogen content of NBR, EPDM, and FKM exposed to 35 and 70 MPa hydrogen pressure immediately after decompression can be predicted using the intercept of the fitted line in Figure 4, and the results are shown in Figure 11a. As the hydrogen pressure increased, the hydrogen content of all rubber materials also increased. NBR with a high solubility had the highest hydrogen content, and FKM with a low solubility had the lowest. Thus, it can be determined that the hydrogen content of rubber materials is closely related to the solubility coefficient. The volume of all rubber materials exposed to 35 MPa hydrogen pressure swelled similarly, but for FKM exposed to 70 MPa hydrogen pressure its volume dramatically increased unlike for NBR and EPDM. Interestingly, FKM significantly swelled in volume despite its low hydrogen content. Meanwhile, NBR with a high hydrogen content exhibited the smallest expansion. Generally, the volume swells to a greater extent as the solute gas content increases the inside rubber material [17]. To investigate the large volume swelling of FKM despite the low contents of hydrogen, the secant modulus of NBR, EPDM, and FKM were compared, as shown in Figure 11b. From this result, the secant modulus and tensile strength of FKM were the lowest compared to those of NBR and EPDM, especially, the result of modulus is consistent with the result of volume swelling of rubber materials irrespective of the hydrogen content. In addition, even if a large amount of gas is adsorbed to the rubber material, the volume swelling sometimes can be small. NBR without the filler had not only the low modulus and tensile strength, but also the volume fairly expanded volume upon exposure to high-pressure hydrogen despite its low hydrogen content. However, after the addition of the filler, the modulus and tensile strength of NBR increased, and the volume swelling decreased despite its high hydrogen content [43,44]. Consequently, the volume expansion depends on the mechanical characteristics regardless of the hydrogen content, and the volume of FKM was more swelled because of its low modulus and tensile strength.

The high-pressure hydrogen deteriorated the tensile characteristics, but both volume and tensile strength returned to the original state according to the time elapsed after decompression. Figure 12 presents the relationship between the relative true strength and the volume ratio. The true strength and the volume ratio can be calculated using the results of the tensile test and the dimensions of the tensile specimen before and after exposure to high-pressure hydrogen. The relationship line was fitted according to the following equation [43]:(9)$$\frac{\sigma_{T}}{\sigma_{T_{o}}}={\left(\frac{V_{o}}{V}\right)}^{\alpha }$$
where α is an experimental constant, which was determined to be 1.53, 1.05, and 1.23 for NBR, EPDM, and FKM, respectively.

A nearly linear behavior was observed for EPDM, which means that the tensile strength returned linearly with the volume. However, NBR and FKM indicated nonlinear behavior. This means that the tensile strength returned only marginally, even if the volume rapidly recovered to some extent in the beginning. On the contrary, the tensile strength largely returned even for a small volume recovery at the end. This phenomenon is attributed to the difference in velocities between the volume and tensile strength of rubber materials. EPDM has an almost similar recovery velocity of the volume and tensile strength, whereas the recovery velocity of tensile strength of NBR and FKM is slower than that of the volume. Consequently, it is clear that the true strength has an affinitive relationship with volume, and the tensile characteristics deteriorate not because of cracks but because of volume expansion upon exposure to high-pressure hydrogen. However, the tensile strength does not always decrease with volume expansion because cracks or blisters are sometimes observed in the fracture area of tensile specimens after exposure to high-pressure hydrogen [45].

## 4. Conclusions

In this study, changes in characteristics of NBR, EPDM, and FKM exposed to 35 and 70 MPa hydrogen pressures were investigated to evaluate the effects of high-pressure hydrogen. The results can be summarized as follows:(1)Compared with NBR and FKM, EPDM has high permeability and diffusivity coefficients because of interactions between hydrogen and the matrix. After exposure to high-pressure hydrogen, EPDM with high permeability had a higher velocity of volume recovery and desorption of hydrogen than NBR and EPDM irrespective of the hydrogen pressure.(2)The volume and hydrogen content of all rubber materials exposed to 35 and 70 MPa hydrogen pressure eventually returned to original values. However, it took a long time for a higher pressure, and the behavior of volume recovery was different from the desorption behavior of hydrogen.(3)NBR had the highest hydrogen content and the lowest volume increase. In contrast, FKM had the lowest hydrogen content and the highest volume increase. The hydrogen content was not related to volume swelling. The mechanical properties can be related to the volume swelling because NBR has a highest modulus and tensile strength than KFM.(4)The characteristics of the CS were not distinctly changed by exposure to high-pressure hydrogen, but the tensile characteristics substantially decreased with increasing volume. The CS specimen exposed to 70 MPa hydrogen pressure showed more severe damage than that exposed to 35 MPa hydrogen pressure, but there was no damage in the fracture area of the tensile specimen regardless of hydrogen pressure. Finally, it was revealed that the decrease in tensile characteristics can be ascribed to the volume increase.(5)NBR, EPDM, and FKM were used in this study; the effect of high-pressure hydrogen was relatively large because the amount of filler was small. The rubber materials with high filler content are less affected by high-pressure hydrogen [10,11,41,45]. NBR, EPDM, and FKM showed a tendency to recover the tensile strength after exposure to high-pressure hydrogen, but the tensile strength of the rubber material without fillers was not recover to its original value due to internal cracks and damages [43]. Thus, it can be considered that the addition of a filler is important to have a high resistance to high-pressure hydrogen in the rubber materials.

## Figures and Tables

**Figure 1 polymers-14-02233-f001:**
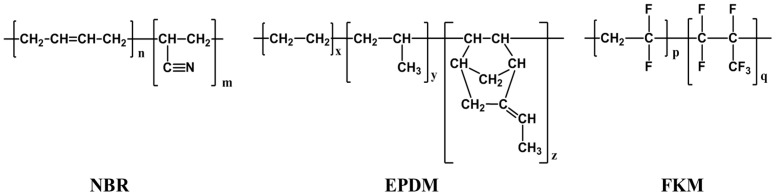
Chemical structure of NBR, EPDM, FKM.

**Figure 2 polymers-14-02233-f002:**
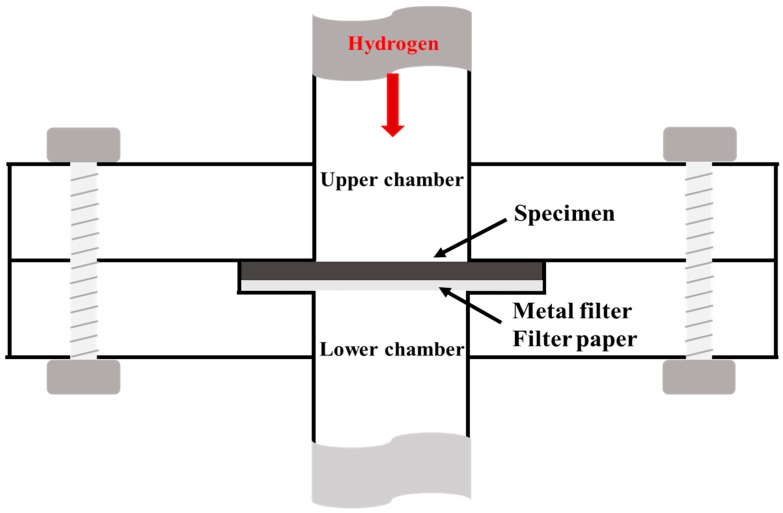
Schematic of the permeation test.

**Figure 3 polymers-14-02233-f003:**
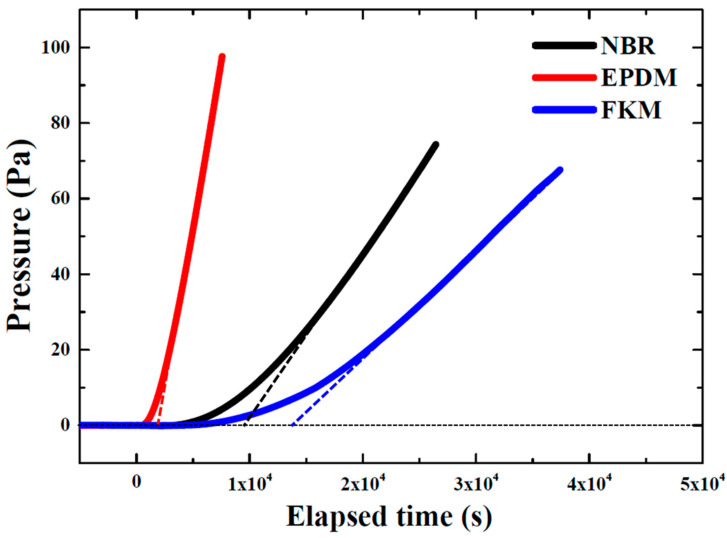
Permeability results of NBR, EPDM, and FKM.

**Figure 4 polymers-14-02233-f004:**
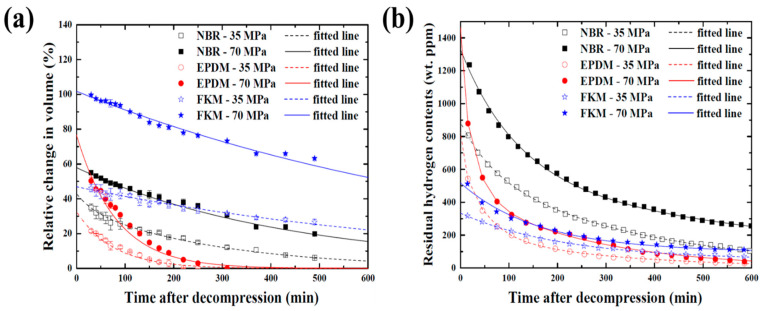
Results of (**a**) volume swelling and (**b**) hydrogen content.

**Figure 5 polymers-14-02233-f005:**
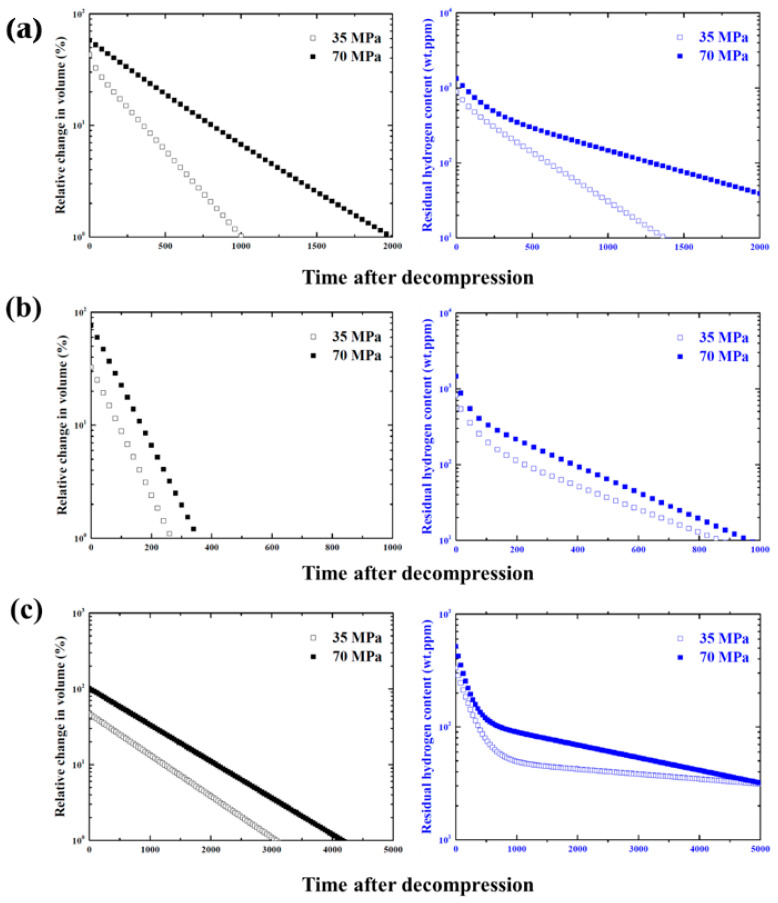
Behavior of volume recovery and desorption of hydrogen. (**a**) NBR, (**b**) EPDM, (**c**) KFM.

**Figure 6 polymers-14-02233-f006:**
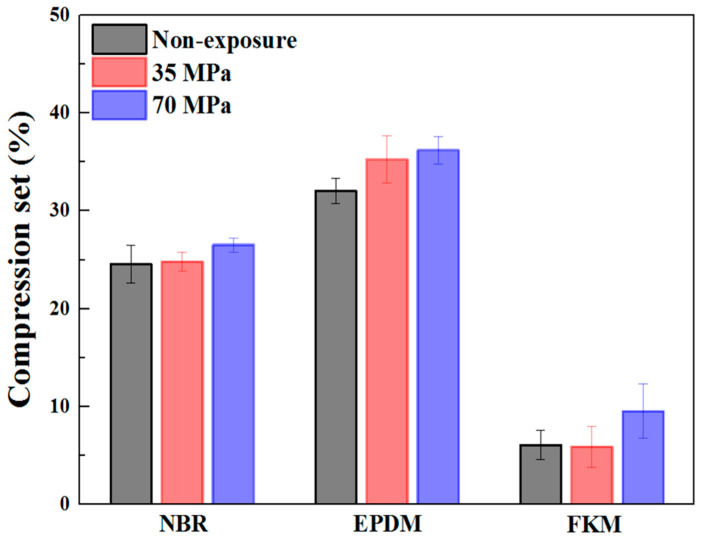
Result of compression set for NBR, EPDM, and FKM.

**Figure 7 polymers-14-02233-f007:**
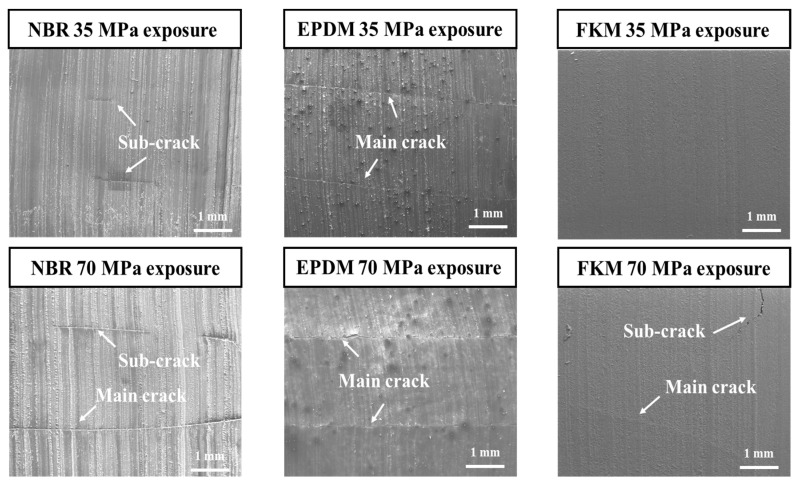
SEM images of compression set for NBR, EPDM, and FKM exposed to 35 and 70 MPa hydrogen pressure.

**Figure 8 polymers-14-02233-f008:**
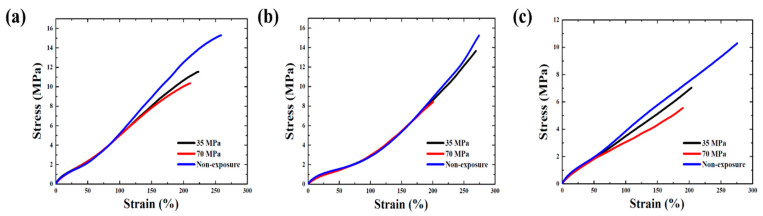
Stress-strain curves of (**a**) NBR, (**b**) EPDM, (**c**) FKM.

**Figure 9 polymers-14-02233-f009:**
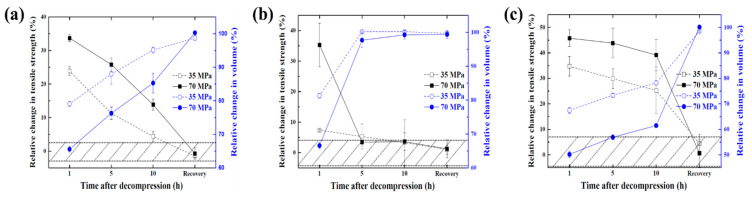
Relative change in tensile strength and volume over time after decompression. (**a**) NBR, (**b**) EPDM, (**c**) FKM.

**Figure 10 polymers-14-02233-f010:**
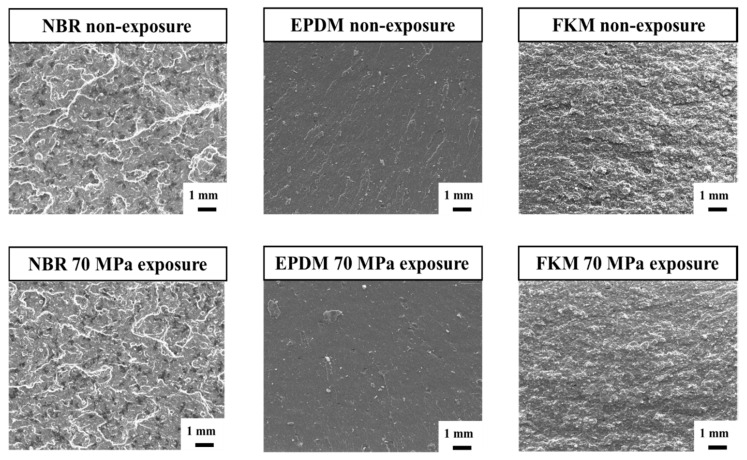
SEM images of fracture area for NBR, EPDM, and FKM not exposed and for 70 MPa hydrogen pressure.

**Figure 11 polymers-14-02233-f011:**
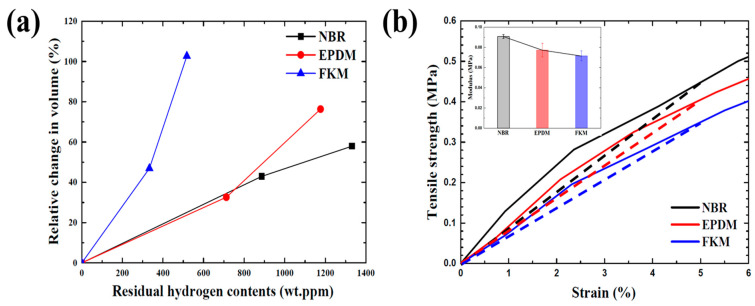
(**a**) Volume swelling and hydrogen content of NBR, EPDM, and FKM and (**b**) tensile modulus of NBR, EPDM, and FKM.

**Figure 12 polymers-14-02233-f012:**
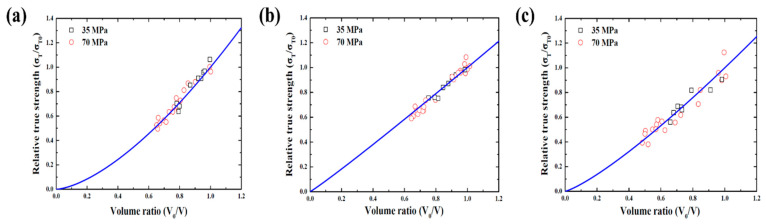
Relationship between relative true strength and volume ratio. (**a**) NBR, (**b**) EPDM, (**c**) FKM.

**Table 1 polymers-14-02233-t001:** Formulation of NBR, EPDM, and FKM (phr).

Ingredient	NBR	EPDM	FKM
Rubber	100	100	100
Carbon black	30 (FEF)	30 (FEF)	25 (MT)
Sulfur	2	-	-
DCP	-	2	-
Ca(OH)_2_	-	-	3
TMTD	1.5	-	-
ZnO	3	5	-
MgO	-	-	4
Stearic acid	1	1	-
DOA	3	-	-
Paraffinic oil	-	5	-
Density (g/cm^3^)	1.3	1.15	1.89
Crosslink density $(\times 10^{-4}$, mol/g)	4.23	5.49	0.63

**Table 2 polymers-14-02233-t002:** Hydrogen permeation parameters of NBR, EPDM, and FKM.

Material	Permeability Coefficient 10^−9^ (mol/m∙s∙MPa)	Diffusivity Coefficient 10^−10^ (m^2^/s)	Solubility Coefficient (mol/m^3^∙MPa)
NBR	3.8	0.9	42.8
EPDM	17.0	5.2	33.4
FKM	2.1	0.8	26.4

## Data Availability

Not applicable.

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
