# Peer review of "Investigation of Physical and Mechanical Characteristics of Rubber Materials Exposed to High-Pressure Hydrogen"

_polymers, 2022, doi:10.3390/polym14112233_

Round 1
Reviewer 1 Report
This manuscript is organized well, and the content is well written. There is very few work focusing on the properties of the rubber materials exposing under high-pressure hydrogen. I think this work is of great interest to the readers.
Minor revision is needed as follows:
- For the stress-strain curves, the Y-axis should be stress rather than tensile strength. Because tensile strength is defined as the stress at the break strain (elongation at break).
- Figure 4, the aspect ratio seems unusual and should be modified.
Reviewer 2 Report
The manuscript is interesting and provides very useful information in the field of elastomeric gaskets. Although there are some details that I would like to clarify.
In the Experimental section, I miss a subsection on the manufacturing process of the different samples (curing temperature, pressure, etc..).
In table 1, what function does Ca(OH)2 in FKM.
How does the filter paper act in the permeation test?
In the Mechanical test section, the compression test is not very clear.
In the tensile test, the H2 Charge test is done with the dumbbell shape or the test is previous to cut.
In the section on permeation of H2, it is necessary to improve the discussion on the difference in solubility between samples.
In the Mechanical Characteristics section, the explanation of the CS is confusing and needs improvement.
The authors need to explain the anomalous behavior of FRM related to increase of volume.
Reviewer 3 Report
It is necessary to investigate rubber materials exposed to high-pressure hydrogen to ensure operational safety. In this study, permeation, volume swelling, hydrogen content, and mechanical characteristics of acrylonitrile butadiene rubber (NBR), ethylene propylene diene monomer (EPDM), and fluorocarbon (FKM) samples exposed to pressure of 35 and 70 MPa were investigated. The investigations are interesting and could be published after revision.
- Volume swelling, hydrogen content, and mechanical tests were performed after exposure to hydrogen pressure of 35 and 70 MPa. Why these values of pressure were chosen ?
- - The authors write that compound formulas are presented in Table 1 ? Chemical structures of NBR, ethylene propylene diene monomer (EPDM) and fluorocarbon (FKM) should be demonstrated.
- -The tested specimens were placed in a high-pressure hydrogen vessel. It seems that the test is different from industrial conditions where the rubber is influenced by the high-pressure hydrogen just from one side ?
- - NBR and EPDM formed a cross-linked structure with sulfur and peroxide, and the compounds were prepared using various rubber additives, such as vulcanization accelerators, co-crosslinking agents, and plasticizers. The process should be described in detail. Chemical structures of cross-linking should be demonstrated.-It is not clear if pure NBR, EPDM, and FKM were exposed to 35 and 70 MPa hydrogen pressures, or cross-linked structures were investigated ?
- -The obtained results should be compared in conclusions with that of other rubber products, which are used in this field of applications and are described in literature.
Round 2
Reviewer 2 Report
The answers provided by the authors are rigorous and conform to my comments.Reviewer 3 Report
If editor and other reviewers agree I could also recommend the paper for publication after the revision.